# Solar Radiation Modification challenges decarbonization with renewable solar energy

Susanne Baur[1], Benjamin M. Sanderson[2], Roland Séférian[3], Laurent Terray[1]

[1]CECI, Université de Toulouse, CERFACS, CNRS, Toulouse, France

[2]Centre for International Climate and Environmental Research (CICERO), Oslo, Norway

[3]CNRM, Université de Toulouse, Météo-France/CNRS, Toulouse, France

*Correspondence to*: Susanne Baur (susanne.baur@cerfacs.fr)

**Abstract.** Solar Radiation Modification (SRM) is increasingly being discussed as a potential tool to reduce global and regional temperatures to buy time for conventional carbon mitigation measures to take effect. However, most simulations to date assume SRM as an additive component to the climate change toolbox, without any physical coupling between mitigation and SRM. In this study we analyse one aspect of this coupling: How renewable energy (RE) capacity, and therefore decarbonization rates, may be affected under SRM deployment by modification of photovoltaic (PV) and concentrated solar power (CSP) production potential. Simulated 1-hour output from the Earth System Model CNRM-ESM2-1 for scenario-based experiments are used for the assessment. The SRM scenario uses Stratospheric Aerosol Injections (SAI) to approximately lower global mean temperature from the high emission scenario SSP585 baseline to a moderate emission scenario SSP245. We find that by the end of the century, most regions experience an increased number of low PV and CSP energy weeks per year under SAI compared to SSP245. Compared to SSP585, while the increase in low energy weeks under SAI is still dominant on a global scale, certain areas may benefit from SAI and see fewer low PV or CSP energy weeks. A substantial part of the decrease in potential with SAI compared to the SSP-scenarios is compensated by optically thinner upper tropospheric clouds under SAI which allow more radiation to penetrate towards the surface. The largest relative reductions in PV potential are seen in the northern and southern hemisphere mid-latitudes. Our study suggests that using SAI to reduce high-end global warming to moderate global warming could pose increased challenges for meeting energy demand with solar renewable resources.

## 1 Introduction

With a rapidly dwindling remaining carbon budget for the Paris 1.5 °C temperature goal, a growing set of literature has been investigating the potential of temporarily reducing climate change impacts with Solar Radiation Modification (SRM), also known as solar geoengineering (UNEP, 2023). SRM is a term describing a set of technologies which temporarily cool the

climate by modifying the balance of incoming versus outgoing radiation (Boucher et al., 2013; Budyko 1977; Crutzen 2006; Irvine et al., 2016). Proposed methods include changing properties of high-lying (Mitchell & Finnegan, 2009) or low-lying (Latham, 1990) clouds, injecting reflective aerosols in the stratosphere (Boucher et al., 2013) or placing objects in space that deflect some of the sun's radiation that reaches the Earth's surface (Angel, 2006). The idea behind SRM is to use one or a combination of these technologies until emissions have been sufficiently reduced and carbon removal technologies scaled up

to keep temperatures at an acceptable level without SRM. This "buying time"-approach hinges critically on the premise of decarbonization during SRM in which a carbon-free energy supply from renewable sources plays an essential role. However, while there is a growing number of studies on the socio-economic effect of SRM on mitigation (Belaia et al., 2021; Bellamy et al., 2016; Burns et al., 2016; Keith, 2000; McLaren, 2016; Merk et al., 2016; Moreno-Cruz, 2015; Wibeck et al., 2015), the so-called moral hazard risk, so far, few studies have assessed whether SRM affects our decarbonization potential in physical

terms. Here, we target the question of changes in renewable energy (RE) generation potential in the case of solar-based RE technologies, photovoltaic (PV) and concentrated solar power (CSP) under SRM.

    A change in solar RE productivity is especially interesting since most SRM methods act on reducing incoming energy, while PV and CSP use incoming energy to turn into electricity. The reduction in solar radiation through SRM is therefore believed

to have negative effects for solar-RE power production (Robock, 2008; Robock et al., 2009). A pronounced effect is expected for CSP under stratospheric aerosol injection interventions, since CSP relies on direct shortwave radiation for its energy production, but the addition of aerosols shifts the ratio of direct and diffuse radiation to entail a larger diffuse fraction. PV panels on the other hand can convert both direct and diffuse shortwave radiation into electrical energy with similar efficiency depending on the specific PV technology used (Parretta et al., 2003) and the tilt of the installation (Khan et al., 2022). However,

while solar RE electricity generation depends substantially on solar irradiance, other atmospheric variables such as temperature and wind influence PV panel and CSP efficiency.

    Most studies to date have estimated global and regional potential of solar-RE and change thereof as a result of global warming. Climate change influences solar energy resources through changes in atmospheric water vapor content, cloudiness and cloud characteristics (Schaeffer et al., 2012; Clarke et al., 2022; Scheeler & Fiedler, 2023) and aerosols (Clarke et al., 2022; Scheeler

& Fiedler, 2023), i.e. atmospheric transmissivity. At a global scale, solar resources are projected to decrease slightly compared to current values (Huber et al., 2016; Crook et al., 2011; Wild et al., 2015) and, in general, power output from CSP plants seems to be more sensitive to climate change than PV plants (Huber et al., 2016). However, electricity production from PV and CSP is not just driven by solar resources, but also by other factors such as surface air temperature and aerosols (Schaeffer et al., 2012; Clarke et al., 2022; Scheeler & Fiedler, 2023). Regional differences in the development of these variables cause

variations in electricity output trends around the world (Scheeler & Fiedler, 2023). Increases in PV and CSP output are projected across Europe, Eastern US and East Asia (Crook et al., 2011; Gernaat et al., 2021; Tobin et al., 2018; Wild et al., 2015; Zou et al., 2019); Whereas Africa, Saudi-Arabia, Australia and Central and South-East Asia show decreases (Crook et al., 2011; Gernaat et al., 2021; Wild et al., 2015). However, not all studies agree on the sign of changes and especially regional

analyses show much greater variability (Bartók et al., 2017; Bazyomo et al., 2016; Huber et al., 2016; Jerez et al., 2015; Tobin et al., 2018).

So far, to our knowledge, only two studies have been conducted on solar RE and SRM. Murphy (2009) investigated the effect of stratospheric aerosols on CSP production concluding that an enhancement of the aerosol layer would lead to a reduction in the efficiency of CSP systems by 4 to 10 % for each 1 % reduction in total sunlight reaching the earth (Murphy, 2009). More recently, Smith et al. (2017) studied PV and CSP potential under stratospheric sulphate injections and found an overall reduction in CSP output of 4.7 to 5.9% on land relative to a scenario without geoengineering and a reduction in PV potential of 1 to 3 % over land depending on the model and type of PV technology (Smith et al., 2017).

In this study we calculate and compare PV and CSP potential under a geoengineered world versus one that is moderately ambitiously mitigated to approximately the same global mean temperature (SSP245) and also versus an unmitigated, fossil-fuel intensive climate change scenario (SSP585). The method of SRM considered in this study is stratospheric aerosol injections (SAI).

## 2 Methods

### 2.1 PV and CSP potential

In this paper we use the term *potential* to refer to an enhanced version of the standard definition of the technical potential. The technical potential is defined as the theoretical potential, the upper limit based on geophysical conditions i.e., the total energy from solar irradiance on Earth, constrained by geographical and technical restrictions. Geographical constraints restrict the theoretical potential to areas that are considered to be physically and regulatorily suitable for PV or CSP production, while technical restrictions refer to efficiency losses during the transformation of primary energy flux to secondary resource (de Vries et al., 2007). Here, we add an additional geographical constraint to the technical potential by weighting the area cells according to the proximity to highly populated cells (see 2.2 geographical restrictions).

### 2.1.1 Data and simulations

We calculate the potential for three different scenarios: SSP245, a scenario representing approximate current policy (O'Neill et al., 2016), SSP585, a very high-emission scenario (O'Neill et al., 2016), and G6sulfur, an SRM scenario that imitates stratospheric aerosol injections (SAI) (Kravitz et al., 2015) and will be referred to as SAI in this study. G6sulfur has the starting conditions and underlying emissions of SSP585 but uses SAI to match the global radiative balance of SSP245 until 2100. G6sulfur is part of the GeoMIP protocol (Kravitz et al., 2015), but here, the setup is enhanced with higher frequency output and additional variables related to radiation and wind. We run the scenarios using the Earth System Model CNRM-ESM2-1 with prescribed aerosol optical depth derived from the GeoMIP experiment G4SSA (Tilmes et al., 2015) to simulate the aerosol

injections in G6sulfur/SAI. 3-member ensembles of G6sulfur/SAI, SSP245 and SSP585 from CNRM-ESM2-1 exist already, but are not used here. Instead, for this study, we repeated the simulations with an alternative version of CNRM-ESM2-1 (Séférian et al., 2019) that accounts for the aerosol-light interaction. This additional feature of the model enables a change in the partition of direct and diffuse light due to a change in aerosol concentration in the whole atmospheric column. We run a 6-member ensemble with initial condition perturbations as for the standard SSP-simulations for all three scenarios in concentration-driven mode. The simulations cover the 2015-2100 period and output data is saved at hourly frequency. The global mean aerosol optical depth required in the SAI simulation to get from SSP585 to SSP245 reaches 0.35 in the last decade. We bilinearly regrid the model output to match the resolution of the land-use data described in Section 2.2. Region allocation for the regional analysis follows the IPCC sixth assessment report working group I reference set (Iturbide et al., 2020), which distinguishes between 46 land regions. In this study, we exclude East and West Antarctica. The calculation of the cloudy sky radiation involves the subtraction of clear sky downwelling shortwave radiation from the total downwelling shortwave radiation. The clear sky downwelling shortwave radiation is a variable that excludes the effect of clouds but includes aerosols, and is saved by the model. To account for the solar geometry for the fixed tilt PV panel configuration, the 1h solar zenith $\theta_z$ and solar azimuth $\alpha$ angles were calculated offline.

## 2.1.2 Photovoltaic potential on horizontal plane

The technical potential for PV ($PV_{TP}$) for grid cell i (kWh/yr) is calculated in line with previous studies (e.g. Dutta et al., 2022; Gernaat et al., 2021; Köberle et al., 2015; Scheele & Fiedler, 2023) as follows:

$$PV_{TP_i} = \frac{RSDS_i}{1000} \times h \times A_i \times a_i \times n_{LPV} \times n_{PV} \times PR \left[\frac{kWh}{y}\right] \qquad (1)$$

With $RSDS_i$ being the shortwave downwelling radiation at the earth's surface, $h$ the hours in a year, $A_i$ the suitability factor for the grid cell, $a_i$ the area of the grid cell, $n_{LPV}$ the land use factor, meaning the area covered by panels, $n_{PV}$ the PV panel efficiency and $PR$ the performance ratio that expresses the difference between performance under Standard Test Conditions (STC) and the actual output of the system due to losses from suboptimal angles, dust and dirt or cable and inverter losses (85%) (Fraunhofer ISE, 2023). The conditions under the STC are a cell temperature of 25°C, solar irradiance of 1000 Wm$^{-2}$ and an air mass spectrum of 1.5 (AM1.5). A collection of all variables, their descriptions, units and sources is provided in Table S1. Equation (2) accounts for changes in efficiency through climate variables:

$$n_{PV} = n_{Panel} \times \left(1 + \gamma[T_{p,i} - T_{STC}]\right) \qquad (2)$$

We have selected the widely used monocrystalline silicon PV panel as our reference panel (Dutta et al., 2022; Jerez et al., 2015; Sawadogo et al., 2021; Feron et al., 2021) whose 2023 panel efficiency under STC is at 26.8 % ($n_{Panel}$) (Fraunhofer ISE, 2023; NREL, 2023) but may be modified under different temperature conditions (Radziemska, 2003) resulting in $n_{PV}$. $\gamma$ represents the efficiency response of monocrystalline silicone PV panels, $T_{STC}$ is the temperature of the panel under STC, i.e. 25°C, and $T_{p,i}$ is the actual temperature of the panel, calculated as:

$$T_{p,i} = c_1 + c_2 \times T_i + c_3 \times RSDS_i + c_4 \times V_i \ [°C] \tag{3}$$

with $c_1$, $c_2$, $c_3$ and $c_4$ as constants that are described in Table S1. $T_i$ is the surface air temperature, $RSDS_i$ downwelling shortwave radiation and $V_i$ surface wind velocity. The PV cell temperature $T_{p,i}$ can be significantly higher than the ambient air temperature $T_i$ under sunny conditions. Total power output is reduced by 0.37% per 1°C increase in cell temperature for monocrystalline cells depending on its temperature coefficient (Mahdavi et al., 2022; Rahman et al., 2015).

### 2.1.3 Photovoltaic potential with fixed tilt panels

To account for the angle with which the shortwave radiation strikes tilted panels, we conduct an additional PV potential calculation that considers the solar geometry and the partitioning of direct and diffuse radiation, instead of using total horizontal downwelling radiation as in 2.1.2. We follow the approach in Smith et al. (2017) and calculate direct radiation as:

$$RSDS_{dir,i} = \frac{RSDS_i - RSDS_{diff,i}}{\cos \theta_{z,i}} \tag{4}$$

With $RSDS_{dir}$ as direct downwelling shortwave radiation, $RSDS_{diff}$ diffuse downwelling shortwave radiation and $\theta_z$ the 1h mean solar zenith angle.

We consider the inclination of the panels to be equal to latitude, oriented towards the equator and calculate the radiation on the tilted panel $RSDS_{panel,i}$ in line with Smith et al. (2017):

$$RSDS_{panel,i} = RSDS_{dir,i} \times \cos \theta_{panel,i} + \frac{1 + \cos \beta}{2} \times RSDS_{dir,i} \tag{5}$$

$\theta_{panel}$ is the angle with which the direct radiation strikes the tilted panel and $\beta$ the tilt of the panel, i.e., the latitude. $\cos \theta_{panel}$ depends on the latitude, solar zenith and solar azimuth angle and is taken as:

$$\cos \theta_{panel,i} = \cos \theta_z \times \cos \beta + \sin \theta_z \times \sin \beta \times \cos \alpha \tag{6}$$

We calculate $PV_{TP,fixed}$ using equation (1), (2) and (3) but replacing $RSDS$ with $RSDS_{panel}$ in (1) and (3). The calculation with fixed tilt panels is used in Figure 4. All other analyses are based on horizontally aligned panels.

### 2.1.4 Concentrated solar power potential

The technical potential for CSP ($CSP_{TP}$) is (Gernaat et al., 2021; Köberle et al., 2015):

$$CSP_{TP_i} = RSDS_{dir,i} \times h \times A_i \times a_i \times n_{LCSP} \times \frac{n_{CSP}}{FLH_i} \ \left[\frac{kWh}{y}\right] \tag{7}$$

$RSDS_{dir,i}$ is downwelling direct shortwave radiation and $n_{LCSP}$ the land-use factor. The Full Load Hours $FLH_i$ are calculated according to Köberle et al. (2015) as:

$$FLH_i = 1.83 \times RSDS_{dir,i} + 150 \tag{8}$$

And the CSP efficiency corrected for atmospheric variables, $n_{CSP}$, as (Gernaat et al., 2021; Dutta et al., 2022):

$$n_{CSP} = n_R \times (k_0 \frac{k_1 \times (T_f - T_i)}{RSDS_{dir,i}}) \tag{9}$$

With $n_R$ as the efficiency of the Rankine cycle (40%), $T_f$ the temperature of the heat transfer fluid (115°C) and $k_0$ and $k_1$ as constants described in Table S1.

Only areas with a 10-year average daily $RSDS_{dir}$ of minimum 4 kWh m⁻² are taken into account, similar to Köberle et al. (2015), Trieb et al. (2009) and Hernandez et al. (2015).

## 2.2 Geographical restrictions


The installation of PV and CSP is subject to geographical constraints such as regulatory constraints, the prevalent land-use and distance to demand, the restriction to onshore areas and, for CSP, the solar resources described in 2.1.3. Figure 1 shows the convolution of the area weights used in the calculation of the renewable energy potential (see variable Ai in 2.1); The single area restrictions and their weights are displayed in Fig. S1. We remove all areas marked as protected with any status as

characterized by the United Nations Environment Programme (IUCN, 2023) as possible solar power installation sites and weigh areas according to the prevalent land-use and distance to highly populated centres as an indicator for the future existence of transmission lines and demand.

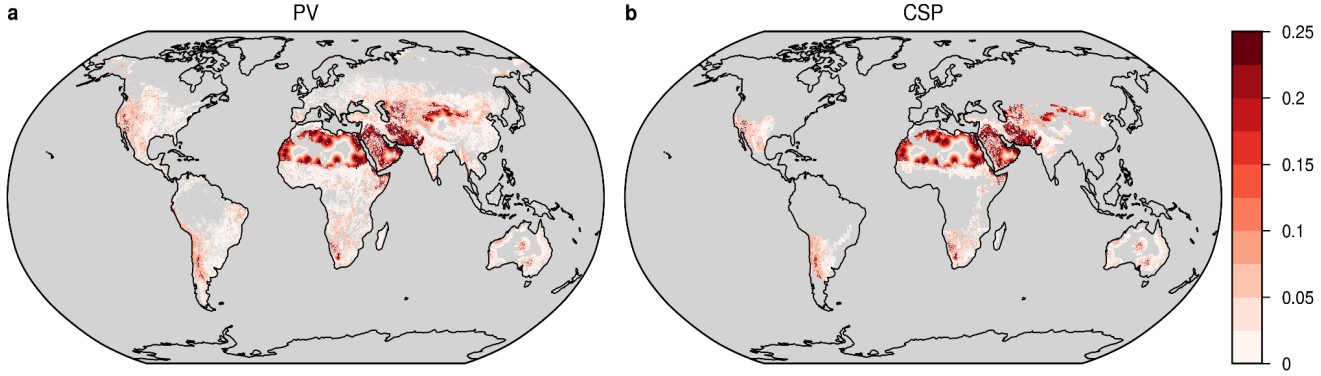

*Figure 1: Area weights applied to a) PV and b) CSP. Weights shown are calculated using SSP2-assumptions.*

Land-use and population density data is taken from the IMAGE3.0-LPJ model (Doelman et al., 2018; Stehfest et al., 2014). IMAGE3.0-LPJ has a spatial resolution of 0.1°x0.1° and distinguishes between 20 different land use and land cover types. We weight each land use and land cover category by the fraction of each grid cell that could be covered by solar renewable energy technology (Ai), similarly to Hoogwijk (2004), but with different land use categories and different fractions assigned (see Table S2 for land use categories and assigned suitability fractions). The underlying idea is that only part of a grid cell is

available for solar RE installations as it may compete with other land uses such as agricultural production or ecosystem services from forests. Another influencing factor is the perceived difficulty of preparing the land for such installations. A suitability fraction of 10% means 10% of the grid cell could be covered with a CSP plant or PV farm. It does *not*, however, imply that 10% of the area is covered by panels, meaning that 10% of the radiative energy can be harvested.

We weight the distance to densely populated areas from the IMAGE3.0-LPJ model (Doelman et al., 2018; Stehfest et al., 2014)

using a sigmoidal function. The data comes in 5-year steps and is aggregated to 10-year means for our calculations. The weight decreases as the distance to the densely populated grid cells increases, until it reaches a weight of 0 at a distance of about 500 km, an arbitrarily chosen cut-off.

The maps and numbers shown in the main paper are calculated using equal weights across all scenarios and years. Please consult the Supplementary Information for figures where area weighting (land-use suitability and population assumption) is

chosen according to scenario (SSP2 for SSP245; SSP5 for SSP585 and SAI). Figure S2 shows the difference in land-use suitability fractions, population density weighting and total area weighting between the scenarios. Figure S3 displays the difference in the same variables for the present (2015-2024) versus the future (2090-2099).

## 2.3 Low Energy Week (LEW)

The Low Energy Week (LEW) metric assesses whether an area shows a change in weekly energy output that is particularly low. This is of interest for energy production as prolonged periods of extremely low production may have greater significance than a slight reduction from high production days to medium production. The LEW metric calculates the number of weeks per year in 2095-2099, where the weekly sum is below the $20^{th}$ percentile of current (2015-2019) seasonal average weekly sums,

$$LEW = \frac{\sum_{i=1}^{y}\sum_{s=1}^{4}(\sum_{j=1}^{h}PVpot_{future}(i,j,s)<\overline{\sum_{i=1}^{y}P_{20}(\sum_{j=1}^{h}PVpot_{present}(i,j,s)))}}{M \times y}, \tag{10}$$

and s indicates the season, y the number of years, h the number of hours per week, $P_{20}$ the $20^{th}$ percentile, M the number of ensemble members and $\underline{x}$ the mean over x. The boundaries of the 7-day period are fixed. Please note that in figures related to LEWs the color bars are reversed compared to the figures related to long-term mean changes. This choice was made because in this case negative values imply a positive outcome for energy production.

## 3 Results

### 3.1 Change in 2090-2099 average PV and CSP potential

On a global scale, the sign of the change in PV potential under SAI compared to SSP245 or SSP585 is negative (Fig. 2a-c). Globally, PV potential is on average 4.1 % lower under SAI than SSP245 and 1.4 % lower under SAI than SSP585. Regionally, this decrease varies from -8 % to 0 % for SSP245 to SAI and -6 % to +4 % in the case of SSP585 to SAI. The largest absolute

losses for SAI are in the northern mid latitudes for SSP585 (-2.5 PWh/year) and the tropical region of the northern hemisphere for SSP245 (-19.0 PWh/year; Fig. 2c). This is likely owing to the large land mass in this latitudinal zone, much of it desert, which is very rich in solar resources. Even small relative losses in this area would be large in absolute terms.

When the area weighting is chosen according to the scenario, the changes between SSP585 and SAI remain the same because assumptions are the same, but for SSP245 there are large differences, mainly related to dissimilar population assumptions (Fig.
S4).

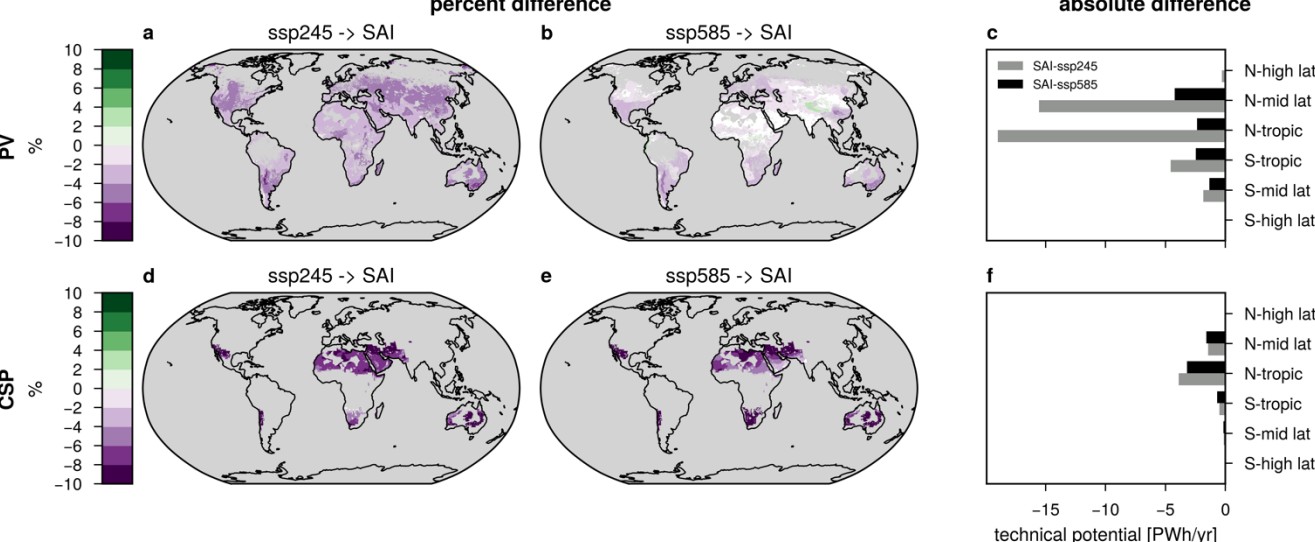

*Figure 2: Difference in 2090-2099 PV (a-c) and CSP (d-f) potential between the ensemble means of SAI and a,d) SSP245, b,e) SSP585 and c,f) absolute difference between latitudinal zonal sums between SAI and SSP245 and SSP585 in PWh/year. White areas have a SNR of < 1. x -> y denotes (y – x)/x.*

The reductions in CSP are greater than those of PV for either scenario comparison. Relative to SSP245, all regions see a decline in CSP potential of 4-16 % under SAI. The comparison with SSP585 shows steeper declines in some areas, such as Australia, where CSP potential is up to 16 % lower under SAI than SSP585. But there are other areas where the decline is smaller than for SSP245, such as in the Middle East. Absolute losses are highest in the northern-hemispheric tropics, mostly due to the large area considered suitable in this latitudinal zone. Globally, CSP potential is reduced by 7.6 % when comparing SAI and
SSP245 and 7.8 % for SAI and SSP585.

Table 1 displays the total global PV potential for each scenario when considering different geographical restrictions (see Table S3 for CSP). Naturally, the potential decreases with increasing geographical restrictions. Irrespective of the constraint applied, SAI has the lowest global potential, followed by SSP585. Unsurprisingly, the potential is significantly reduced by the
weighting of land use suitability. Additionally, the weighting of population density has a notable impact due to the fact that areas with the highest potential, such as deserts, are typically further away from population centres than areas with lower potential such as crops. For CSP, there is only a minimal difference between SSP245 and SSP585, but there is a considerable reduction for SAI. Without land-use suitability or population density weighting SSP585 has the highest potential of the three scenarios (Table S3).

*Table 1: Total global 2090-2099 PV potential per scenario in PWh/yr under different geographical constraints.*

| Geographical constraints | SAI | SSP585 | SSP245 |
|---|---|---|---|
| Land areas | 35,391 | 35,859 | 36,903 |
| Unprotected areas on land | 29,733 | 30,129 | 31,014 |
| Unprotected areas on land weighted with suitability fractions | 1,686 | 1,701 | 1,754 |
| Unprotected areas on land weighted with suitability fractions and distance to highly populated areas | 1,044 | 1,054 | 1,085 |

Figure 3 explains which physical variables drive the change in PV potential and create the regional variation visible in Fig. 2. When holding radiation fixed and letting temperature be the only variable that fluctuates, the difference in PV potential is very small, especially for SSP245 and SAI, where global mean temperature is at a close match (Fig. 3a-c). The regional variation change is a result of differences in surface radiation (Fig. 3d-f), with changes in cloudiness being the primary driver of variance (Fig. 3g-l). With few exceptions, the cloudy sky in SAI is much more transmissive for shortwave radiation, increasing the potential by up to 16 % for cloudy sky. Compared to SSP585, cloud cover is enhanced under SAI in some areas such as over large parts of Australia, South-East Africa, Northern Argentina and Southern United States and Mexico. However, despite SAI having fewer reflective clouds than SSP245 and SSP585, the reduction in clear-sky radiation is more pronounced. With regard to radiation, the potentials are significantly more negative when comparing to SAI than when the two SSP-scenarios are compared to each other (Fig. 3g-i). This is because the principle of SAI is the reduction of incoming solar radiation. See Fig. S5 for the physical drivers of CSP.

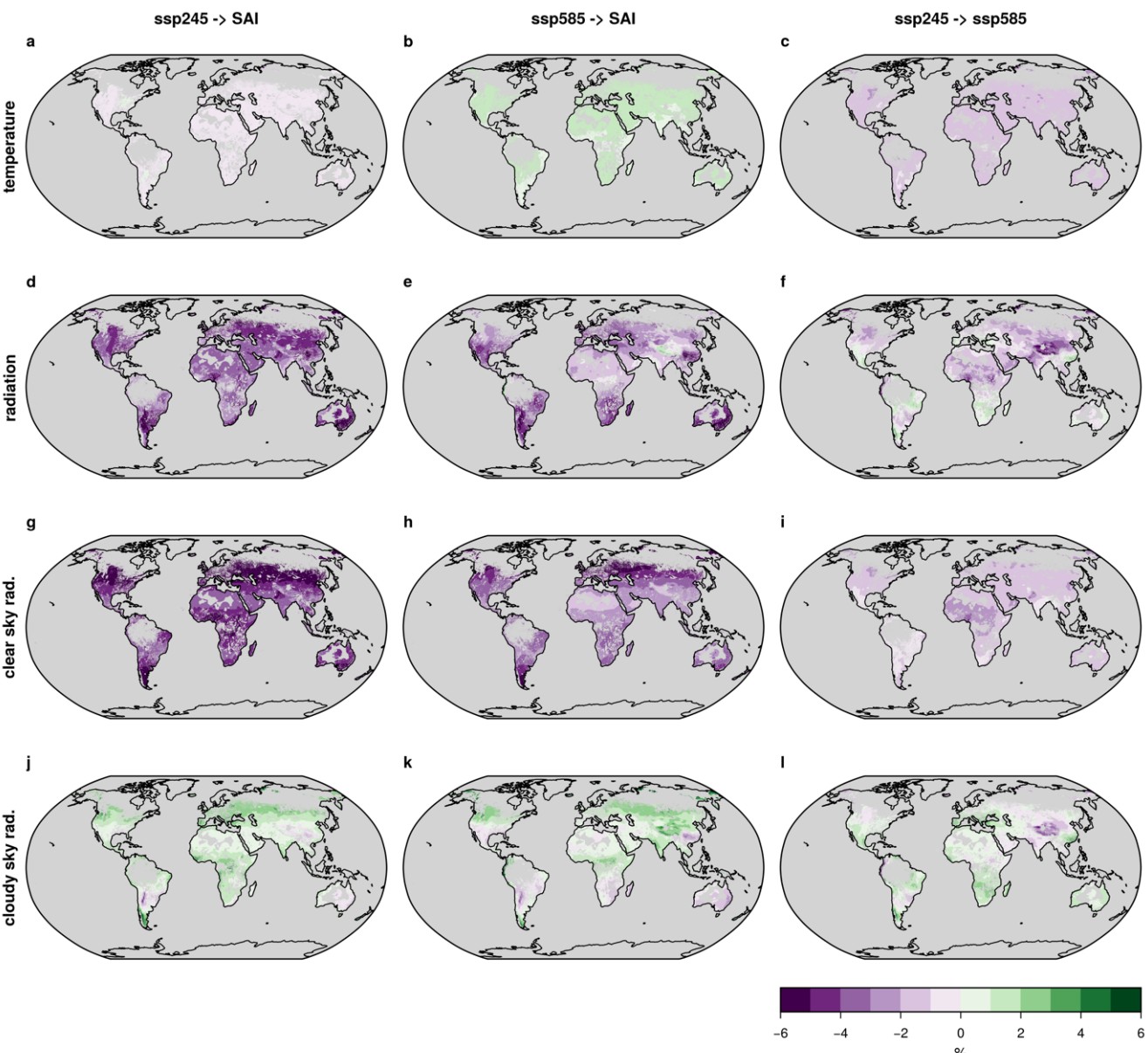

*Figure 3: Main drivers of change in 2090-2099 PV potential, a-c) surface air temperature, d-f) total downwelling surface radiation, g-i) clear-sky radiation and j-l) cloudy-sky radiation. Areas with SNR < 1 are shown in white. x -> y denotes (y − x)/x. a-c) was calculated by keeping all variables except temperature fixed, d-f) by keeping all variables except radiation fixed, g-i) by using the model output clear sky radiation instead of total radiation (see Methods) and j-l) by subtracting clear sky radiation from total radiation.*


### 3.2 Change in 2090-2099 average PV potential with fixed tilted panels

Figure 4 displays the same elements as Figure 2 but under consideration of the sun's position to the tilted panels. Compared
to the calculation with horizontal panels, on average, more direct radiation reaches the panels and less diffuse radiation (Figure
S6). This leads to more total radiative energy reaching the panels in many latitudes but particularly in the mid-latitudes and
especially for the SSP scenarios. Because this effect is more pronounced under the SSP-scenarios than SAI, it further increases
the relative and absolute difference between SAI and SSP in these higher latitudes compared to the calculation with horizontal
panels (Fig. 2, 4), shifting the biggest absolute losses for SAI compared to SSP245 to the Northern mid-latitudes instead of the
tropics. With fixed tilt panels, total global PV potential is 6.9 % lower under SAI compared to SSP245 and 4.2 % lower
compared to SSP585. Reductions under SAI are most pronounced over Russia and Central Asia, the Great Plains in North-
America and Argentina in South America, New Zealand and south-east Australia.

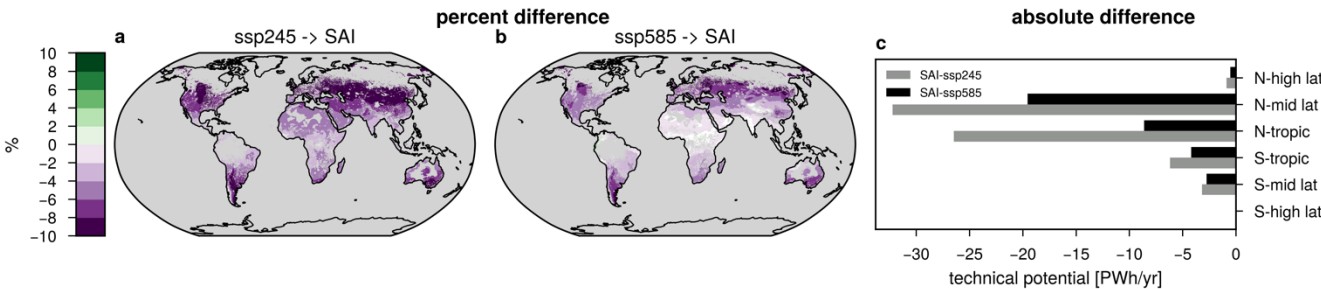

*Figure 4: Difference in 2090-2099 PV potential with fixed tilted panels between the ensemble means of SAI and a) SSP245, b) SSP585 and
c) absolute difference between latitudinal zonal sums between SAI and SSP245 and SSP585 in PWh/year. White areas have a SNR of < 1. x
-> y denotes (y – x)/x.*

### 3.3 Changes in potential on regional scale and with higher frequency

The following analysis looks at changes in PV potential on more refined spatial and temporal scales. Figure 5a presents relative
regional changes compared to the SSP-scenarios split up into two different seasons: December, January, February (DJF) and
June, July, August (JJA). The smallest variation in seasonal potential is in the Sahara, the largest in Central North-America for
SSP245 and Southern Australia for SSP585. Most regions see the largest decrease in potential in their respective hemispheric
winter. Figure S7 and S8 display all 44 regions (Fig. S7, S8). When including the area weighting of the respective scenarios,
some regions experience a substantial shift and in the case of north-west North-America, western North-America, the
Caribbean, South Asia and southern Australia even a change in the sign of difference (Fig. S9 for land-use suitability difference
only; Fig. S10 for land-use suitability and population density difference).

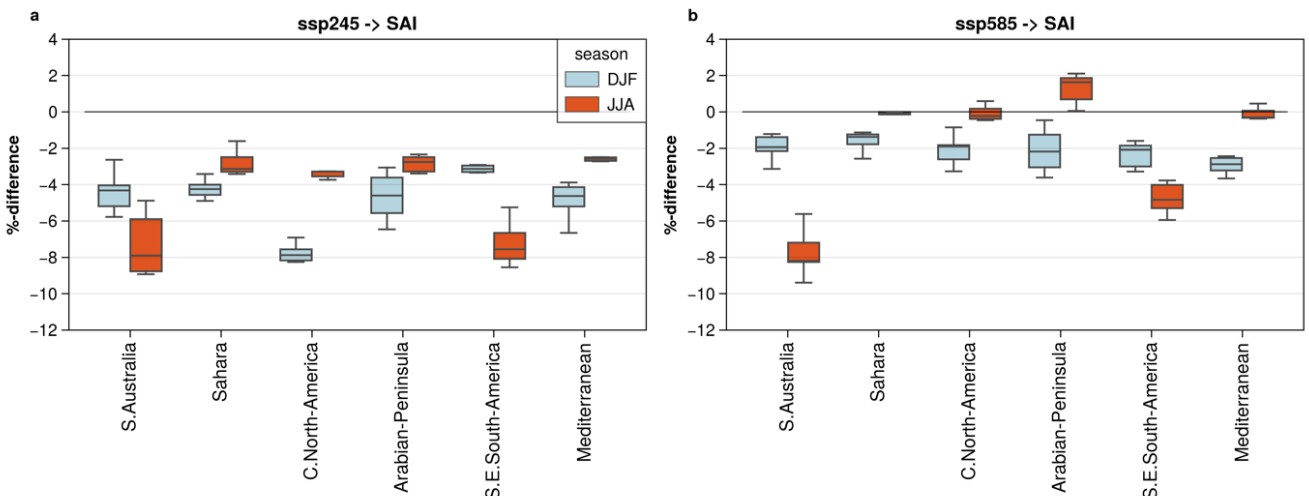

*Figure 5: Relative change in 2090-99 PV potential from a) SSP245 to SAI and b) SSP585 to SAI for 6 IPCC AR6 regions (Iturbide et al., 2020) split up into two seasons of December, January, February (lightblue) and June, July, August (orangered). Boxplot bars represent the spread over the 6 ensemble members. X -> y denotes (y – x)/x.*

Further refining the temporal resolution, the maps a-c in Fig. 6 show the change in PV Low Energy Weeks (LEWs; see definition in 2.3) from present (2015-2019) to future (2095-2099) for the three scenarios SAI, SSP245 and SSP585, when the area weighting is kept constant between the present and the future as well as between the scenarios. Please note that the color bar is reversed in this figure compared to Fig. 2, 3 and 4. We made this choice because in this case negative values imply a positive outcome for energy production. There appear to be similar regional trends among the scenarios, but the extent of

change varies considerably: Under each of the three scenarios, most regions see an increase in LEWs from present to future. Similarly, in all three scenarios, western China, the southern Sahara and central North-America, areas with high absolute potential, see the largest increases in low solar resource periods, with SSP585 experiencing the largest one. Areas that are white, i.e. have up to 10 LEW per year, either experience a decrease in LEW compared to present or remain constant at the present-day value.

SAI has substantially more LEWs than SSP245. Highly productive regions, such as the Saharan and East African region, Tibet and West China, Australia, Madagascar, Northwest Argentina and the Middle East including the Arabian Peninsula are especially affected with up to 12 additional low resource weeks per year (Fig. 6d). The only areas with a small decline in LEWs compared to SSP245 are the South of India, the west of Spain, Portugal and along the Argentinian and Chilean border. Compared to SSP585, SAI clearly benefits several regions by having up to 8 LEWs less per year on the Tibetan Plateau and

up to 4 weeks less in West China, the Arabian Peninsula, northern North-America and parts of the Sahara and Central Africa and Russia (Fig. 6d,e). However, it exacerbates the negative trend in others, adding up to 8 additional LEWs per year (East China, Southern Africa and Madagascar, Southern/Central South-America, Australia and the Middle East excluding the

Arabian Peninsula) (Fig. 6e). Generally, in areas that experience the most pronounced rise in LEWs from present to future under SSP585, SAI offsets some of these increases.

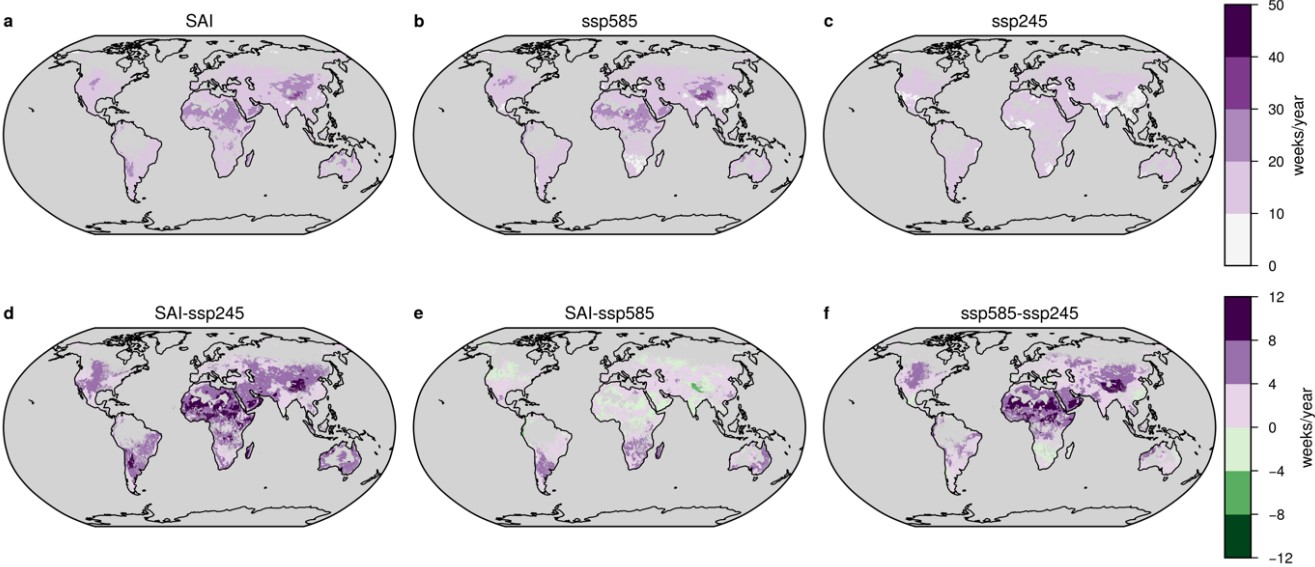


*Figure 6: PV Low Energy Week metric for a) SAI, b) SSP585 and c) SSP245. The LEW is calculated between the present (2015-2019) and the future (2095-2099) with equal area weighting. See 2.3 for the LEW equation. D-f) are the differences between a-c). Color bars are reversed compared to Fig 2,3,4.*

For CSP, even though relative changes are much stronger than for PV, the difference in number of LEWs is comparable to if not less than for PV. Relative to SSP245, the Sahara, the Middle East and Australia register the largest increase of LEWs per year under SAI. Botswana and northern Namibia see a small decrease. The pattern is similar when comparing to SSP585, although here not southern Africa but the Arabian Peninsula is the most advantageous under SAI and shows a decrease in LEWs. The maps d and e in Fig. S11 are confined to areas deemed suitable under SAI and exclude regions that may be
considered under the SSPs but not under SAI. The signal in the maps a, d and e is dominated by the fact that different areas are considered suitable for CSP in the present versus the end of the century.

Over the American continent, trends in 10-year means calculated from hourly data are consistent with weekly sums (Fig. 6d-f; see Fig. S12 present-future comparison). However, elsewhere there are clear differences: for example, one of the areas most heavily affected in relative terms of 10-year average differences from SSP245 to SAI is the south of Australia. But, in terms
of LEW difference between these two scenarios, while SAI clearly records more LEWs in that region than SSP245, it is by no means the area of highest LEW increases.

The shift from present (2015-2024) to future (2090-2099) implies the largest drops in PV potential for SAI (4-8 %), with
SSP585 following closely behind, whereas SSP245 displays only decreases of 0-4 % with certain regions experiencing an
elevated potential (Fig. S12). Figure 7 illustrates the temporal evolution of the relative difference between the three scenarios
and present-day values with the increase in SAI deployment intensity over time. In the first four decades, the scenarios differ
only slightly, but the gap in potential starts to widen as time goes on. The quasi-linear increase in the gap between SAI and the
SSP-scenarios in some regions indicates that, in these areas, the reduction in PV potential strengthens with increasing global
mean aerosol optical depth. See Fig. S13 for the temporal evolution of all 44 regions. The temporal evolutions are a lot less
well-behaved with relative increases and decreases compared to the present that can be several times larger when the land-use
suitability area weighting is included respective to its scenarios (Fig. S14) than when it is not (Fig. S13).

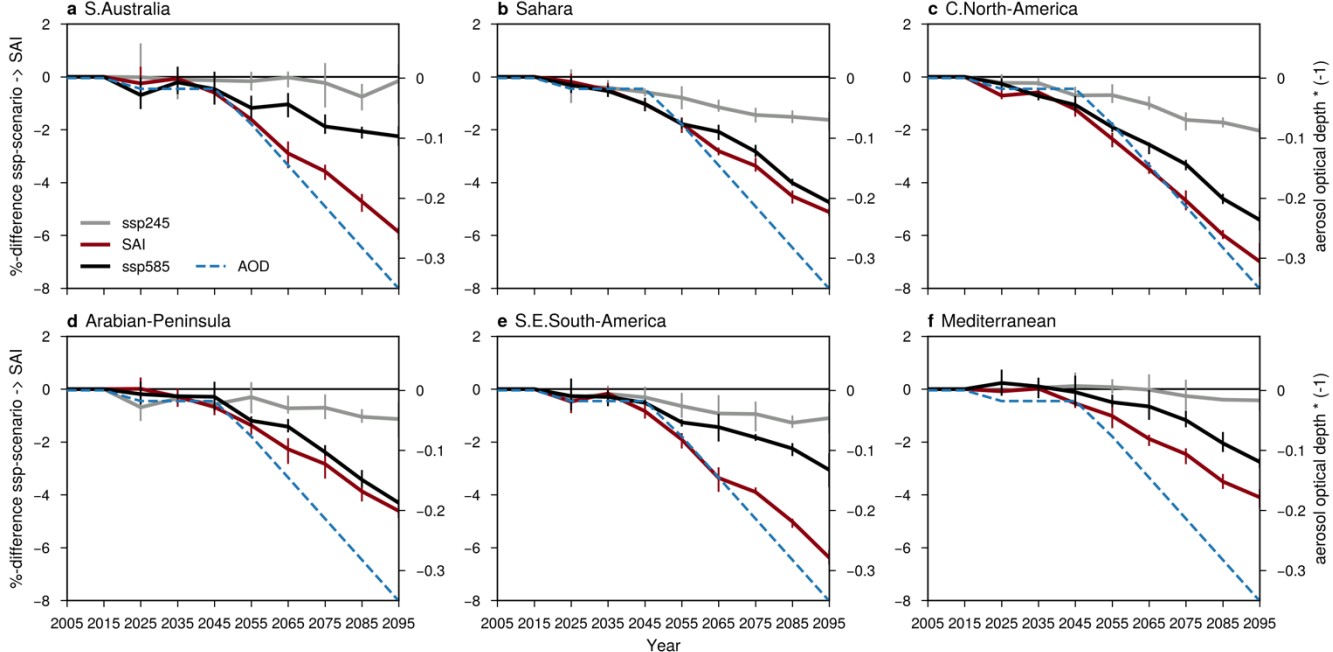

*Figure 7: Relative difference over time of SAI (red), SSP245 (gray) and SSP585 (black) PV potential compared to 2015-2024 values for
selected regions. Lines are the ensemble means with the bars indicating the 20-80 percentile ranges of the single members. X -> y denotes
(y − x)/x.*

## 4 Discussion

Current SRM scenarios rely on the assumption that, in physical terms, the decarbonization potential remains unchanged
whether SRM is added to the toolbox or not. Here, we scrutinized this hypothesis by looking at changes in the potential of
solar RE technology such as photovoltaic (PV) and concentrated solar power (CSP). The data suggests that when comparing
an SRM-world to a world where mitigation was chosen over SRM, i.e., SSP245 in our setup, almost all regions would have

several additional weeks per year with low energy potential under SRM (Fig. 6). With respect to the baseline-scenario SSP585, regions that are especially affected by climate change appear to benefit from SAI and exhibit less LEWs, but the overall trend is nevertheless an increase in LEW. When tilted panels are considered, relative and absolute losses under SAI compared to the SSP-scenarios are much greater, especially in mid and high latitudes (Fig. 2, 4).


In terms of long-term changes the potential is ubiquitously reduced by several percentage points under SAI, especially for CSP (SSP245: -7.6 %) but also for PV (SSP245: -4.1 %; fixed tilt panels SSP245: -6.9 %) (Fig. 2 a,d, 4). When comparing the SAI simulation with SSP585, the reduction in PV potential is still dominant (SSP585: -1.4 %; fixed tilt panels SSP585: -4.2 %) but

less pronounced with few regions showing insignificant differences (Fig. 2 b, 4). In contrast, the change in CSP potential between SAI and SSP585 does not differ that much from the change between SAI and SSP245 (SSP585: -7.8 %; SSP245: -7.6 %) (Fig. 2 d, e). Smith et al. (2017), the only previous study that used a modelling framework to analyse PV and CSP under geoengineering found an average decrease in PV power output on land for fixed panels of 1 to 1.7 % and a decrease of 4.7 to 5.9 % for CSP, depending on the climate model and without confinement to suitable locations. The greater relative losses we

record are likely due to the larger temperature gap we offset with SRM in our experiments. Their study used stratospheric aerosol injections to cool global mean temperature by 1°C from an RCP4.5 starting point, which resembles our SSP245 end point. This means that not only do we have a stronger climate change signal underlying our data, but we also use SRM to cool down more than double the temperature difference that their experiments offset and, as Figure 7 demonstrates for the case of PV, many regions see a quasi-linear decrease in potential the more SRM is scaled up. Other sources of discrepancy between

results might lie in the different implementation of SRM in the climate models, the higher temporal resolution of our output data and the different approach in calculating PV potential. Nevertheless, in line with Smith et al. (2017) we find that the change in solar radiation due to geoengineering has a more profound effect on PV and CSP potential than changes in surface air temperature do (Fig. 3, Fig. S5) and one of the two models they use, GEOSCCM, shows a similar pattern in relative PV potential changes under SAI compared to a control simulation as our results (Fig. 4).


While a decrease in PV and CSP potential was to be expected as the very nature of SRM is to reduce incoming solar radiation, there are two drivers that exert a positive impact of SAI on PV panels compared to the SSP-scenarios for horizontal panel alignment. The first driver is the temperature benefit that photovoltaic panels get from colder ambient air temperatures (Dubey et al., 2013), which we observe for the difference between SAI and SSP585 (Fig. 3b). Surface air temperatures are similar

between SSP245 and SAI. Thus, in this scenario, the advantages of geoengineering on photovoltaic panels in terms of temperature are practically non-existent (Fig. 3a). The second driver is the reduction in reflective cloud cover under SAI. With respect to SSP245 (Fig. 3j), this change compensates a significant part of the decrease in PV potential. For SSP585, while overall reflective cloud cover is also lower under SAI, the difference is less pronounced and the sign of change in cloud cover is region-dependent. Here, areas with substantial reductions in PV potential correlate well with regions where cloud optical

depth is actually enhanced under SAI (Fig. 3 e,k). We are not the first to have observed a change in cloud cover under

geoengineering: Kuebbeler et al. (2012), Cziczo et al. (2019) and Visioni et al. (2018) showed that the injection of aerosols into the stratosphere can affect upper tropospheric clouds, making them optically thinner (Kuebbeler et al., 2012; Cziczo et al., 2019; Visioni et al., 2018). The main drivers of this effect are the reduced vertical temperature gradient in the troposphere which leads to a decreased ice supersaturation probability and thereby a decreased ice particle number density and optical

thickness (Kuebbeler et al., 2012; Visioni et al., 2018) and the aerosols themselves (e.g. Visioni et al., 2021). These optically thinner upper tropospheric clouds allow more shortwave radiation to propagate downwards which leads to the radiation compensating effect seen in Fig. 3j,k. Visioni et al. (2021) illustrate the global pattern of total cloud cover differences between a SAI simulation that compensates for the temperature increase above present values under RCP8.5 conditions and the control run close to the present period. The simulations were performed with the CESM-WACCM model. Not only did they find a

dominant reduction in total cloud cover under the aerosol injections, but, similar to our results, their experiments show increased cloudiness in areas such as Australia, north-west South-America and parts of China. This pattern, although not perfectly identical, correlates well with the cloud effects observed in our results (Fig. 3j,k). The decreased potential under SSP585 compared to SSP245 for clear-sky calculations is likely due to the increase in atmospheric water vapor from climate change as observed by Scheele & Fiedler (2023) that tends to prevent a fraction of shortwave radiation from propagating all

the way to the surface. Contrary to PV, CSP benefits from the warmer temperatures under SSP585 than SAI or SSP245 (Fig. S5). This adds to the negative trend due to radiation changes from SAI compared to SSP585 and compensates some of the negative effects compared to SSP245, leading to a more similar overall reduction between the SSP-scenarios and SAI (Fig. 2 d, e).

Adding the solar geometry and the tilt of the panel to the PV potential calculation is arguably a closer depiction of a real-world application in higher latitudes than a horizontal panel alignment. Due to the almost horizontal alignment of the panels in the tropics, the difference between the two modes of PV potential calculations is minimal in low latitudes. However, tilted panels allow more total radiation to be harvested in higher latitudes than horizontally aligned panels because they catch more direct radiation. The inclination leads to an increase in direct beam on the panel and a decrease of diffuse radiation that reaches the

panel. In most latitudes, the benefit of the increased direct beam outweighs the decrease in diffuse radiation for all three scenarios. We see a larger reduction in PV potential under SAI compared to SSPs in higher latitudes for tilted panels because the aerosols in SAI scatter the incoming radiation, reducing the ratio of direct versus diffuse light and hence reducing the benefit of the tilt. Hence, relative reductions in high latitudes under SAI that already exist for horizontally aligned panels are further increased for tilted panels. Therefore, under a geoengineering intervention, where the partitioning of direct and diffuse

radiation shifts to include a larger diffuse part, tilting the panels, while still advantageous in most latitudes, becomes less useful in maximizing radiative energy.

The total global energy potential of our analysis is broadly comparable with the large range given by existing studies that have looked at technical PV and CSP potential in the present or under climate change and provide their output in energy units

(Hoogwijk, 2004; de Vries et al., 2007; Köberle et al., 2015; Chu and Hawkes, 2020) (Table 2). Our global PV potential is at the upper end also due to the assumptions we make regarding the technological efficiency and socio-economic circumstances at the end of the century.

*Table 2: Comparison of the PV and CSP potential with previous studies. Our results: SSP245. NA means not available.*

|  | PV [PWh/yr] | CSP [PWh/yr] | Year |
|---|---|---|---|
| **Our results** | 1085 | 99 | 2100 |
| **Hoogwijk, 2004** | 366 | NA | 2000 |
| **de Vries et al., 2007** | 939 / 4105 | NA | 2000 / 2050 |
| **Köberle et al., 2015** | 101 | 173 | 2010 |
| **Chu & Hawkes, 2020** | 836 | 587 | 2008-2017 |

The pattern and relative decrease in potential from greenhouse gases and other SSP-scenario inherent climate active substances such as aerosols are broadly aligned with other studies that have analysed PV and CSP potential under climate change (Crook et al., 2011; Wild et al., 2015; Gernaat et al., 2021; Zou et al., 2019; Scheele und Fiedler, 2023). For example, most of the above-mentioned studies report the strongest signal of increase over Europe and East Asia. While our results do not point to a large increase in this area, we do see small increases or only little change over the same areas. The strongest negative signal

in our results is over Tibet, which also broadly agrees with previous studies.

A major challenge with RE is the intermittent nature of the supply, rendering a high temporal resolution of output data a crucial aspect of any RE analysis. LEW is therefore a more relevant metric than 10-year average changes when it comes to energy production. Our Low Energy Week metric suggests that the observed negative trend in 10-year averages for SAI with respect

to SSP245 does not necessarily translate to an increase in extended periods of very low solar resources (Fig. 2a, Fig. 6d). While other studies have found only minor changes in the variability of solar PV energy production (Tobin et al., 2018; Jerez et al., 2015), our results demonstrate that, under geoengineering, the distribution of PV potential is more complex than a simple shift of the mean and standard deviation and it might be important to look at high temporal resolution output.

The reduced productivity from geoengineering would be added on top of the burden of climate change (Fig. 7, S12). Meaning the failure to decarbonize early enough, which would render SRM unnecessary, makes it even more difficult to decarbonize later once SRM is deployed. Reduced solar RE productivity implies higher land-use and financial requirements to generate the same amount of energy. This makes the technology less cost-efficient and less competitive against other sources of energy including fossil alternatives. If fossil alternatives were chosen over solar RE it could prolong the duration of the SRM

deployment, implying higher risks and costs (Baur et al., 2023). If, on the other hand, solar RE is chosen despite the reduced productivity it would imply higher costs due to the greater amount of infrastructure required to generate energy. The results of

this study should therefore be considered when constructing SRM scenarios that assume unchanged technical emission reduction potential under geoengineering.

Indeed, our analysis demonstrated how sensitive results are to assumptions that are independent of the change in resources, such as shifts in population and land-use suitability (Fig. S4, S13). The same conclusion has been drawn by previous studies on RE (de Vries et al., 2007). Attempting to predict these variables with any degree of accuracy is, of course, likely to fail. By including area restriction and weighting in this study, but excluding any variation in these assumptions between the scenarios, we allowed the change in solar resources to be the driver of change while focussing on areas that are of interest for solar energy

parks.

The main drivers of change in this study are the different types of radiation and temperature and Fig. 3 shows their relative importance for PV potential between the scenarios. In reality, these variables are not independent and a breakdown to these single components may not be fully physically correct. However, it provides an idea of which differences between the scenarios

are contributing to the total relative change in PV potential we see in Fig. 2. Fig. 3 demonstrates how variation in cloud properties between the scenarios has a large effect on the result, especially the reduction in reflective cloud cover under SAI compared to SSP245 and SSP585. While other studies have found similar effects under SRM (Kuebbeler et al., 2012; Visioni et al., 2018), the parameterization of clouds in the climate models underlying theirs and our study is highly idealized and should be interpreted with caution. Furthermore, in CNRM-ESM2-1, since there is no interactive sulphur cycle and no

stratospheric aerosol microphysics, SAI is represented by imposing a sulphate distribution that is calculated offline (Visioni et al., 2021).

Other, more local SRM scenarios have been proposed, such as injecting aerosols solely over the Arctic (Jackson et al., 2015) which would have no direct effect on solar power output elsewhere. For a more in-depth discussion of the effects of different

types of SRM on solar power output see Smith et al. (2017).

Finally, while solar renewable energy sources will likely play a role in the net-zero transition, they are only one part of it. Effects on other renewable sources such as wind, hydropower and bioenergy have to be considered to be able to draw a conclusion regarding the full effect of SRM on decarbonization, as well as an in-depth analysis of the effects on the carbon

cycle. Furthermore, it may not be a question of a small reduction in solar RE productivity that is decisive, but rather whether the infrastructure for PV and CSP is actually developed on a significant scale. This depends on much more than just the technical potential. Production costs, the production costs and availability of alternatives, policy incentives like subsidies and feed-in tariffs etc. all play a major role when it comes to the choice and actual implementation of the energy generating system.

## 5 Conclusion

SRM is increasingly being considered as a tool to supplement traditional mitigation measures in reducing anthropogenic climate change. Currently, such simulations assume no physical link between SRM and the ability to decarbonize. Here, we assess one aspect of this coupling: whether SRM affects the potential to reduce emissions through its impacts on solar renewable energy.

We find that there is a significant decrease in PV and CSP potential under SAI compared to SSP245, both in terms of number of weeks in a year with low solar resources and in terms of 10-year means. Compared to the baseline-scenario SSP585, SAI appears to counterbalance some adverse effects in regions that see especially pronounced reductions in PV potential due to climate change, but overall, it worsens the trend. Regarding CSP, the increase in LEWs is of a similar magnitude to that of PV, but the 10-year average trends show twice the decline in potential for CSP than for PV.

The 10-year average trends in our results support existing theories (Murphy, 2009; Robock, 2008; Robock et al., 2009; Smith et al., 2017). However, our study adds a more comprehensive quantitative aspect to the discussion, especially with regards to the greater temporal resolution of our findings. Indeed, despite certain similarities in the trends of the 10-year averages and the weekly sums of hourly output (LEW), we note that there are regions, including some with high absolute potential such as desert areas, that show different developments under the 10-year mean changes and the weekly sums. When a tilt of the PV

panel is taken into account, the reductions in potential under SAI become even more pronounced at higher latitudes. This is because the inclination maximizes the amount of direct radiation reaching the panel, while SAI alters the ratio of direct and diffuse radiation to increase the diffuse part. Therefore, when SAI is used, tilting the panels, while still advantageous in most latitudes, becomes less useful in maximizing radiative energy.

Since the principle of SAI is to cool temperatures by reducing incoming radiation, a reduction in potential was to be expected. However, there are two drivers we identify that change the outcome of PV in favour of SAI: Changes in upper tropospheric cloud cover compensate for a substantial amount of the decreased potential under SAI compared to both, SSP245 and SSP585. As previous studies have suggested, this change in high altitude cloud cover is mainly the result of temperature anomalies in the lower stratosphere due to the aerosols (Kuebbeler et al., 2012; Visioni et al., 2021, 2018). For SSP585, the reduction in

regional temperatures is beneficial for photovoltaic panels, increasing their productivity, but disadvantageous for CSP.

Reducing emissions is one of the greatest challenges facing society today, and the record-breaking greenhouse gas emissions (Friedlingstein et al., 2022) indicate that we are still grappling with implementing effective mitigation strategies (UNEP, 2022). One of the key approaches that can take us closer to achieving net-zero is the extensive implementation of sustainable energy

sources such as PV and CSP (IPCC, 2011, 2022). Previous research has demonstrated (see for example Huber et al., 2016; Crook et al., 2011; Wild et al., 2015; Clarke et al., 2022; Scheele & Fiedler, 2023), and this study supports these findings, that

climate change itself can already lead to a reduction in PV output in some regions. Here, we demonstrate that SRM would increase the challenge of mitigation further by reducing PV and CSP potential. This is another argument for early mitigation, as it suggests that failing to decarbonise early enough, which would render SRM unnecessary, makes it even more challenging to decarbonise later when SRM is implemented. And, since net-zero greenhouse gas emissions is a crucial component of the buying-time approach of SRM, this study's findings suggest that such an approach may be more challenging than previously recognized. This should be factored into the construction of SRM experiments that currently assume no coupling between SRM and mitigation. Of course, solar RE is only one part of the overall mitigation strategy and additional research is needed to establish a full picture of the linkage between SRM and decarbonization. This requires a complete and comprehensive assessment of other physical and ecological couplings between mitigation processes, SRM and carbon-climate feedbacks.

**Code and data availability statement**

The code and postprocessed data is available at https://doi.org/10.5281/zenodo.10658589 (Baur, 2024).

**Author Contribution**

All authors conceptualized the study and SB carried it out. SB ran the simulations and prepared the manuscript with contributions from all authors.

**Ethics Declaration**

One of the co-authors is on the editorial board of Earth System Dynamics. The authors declare no other conflicts of interest.

**Acknowledgements**

Susanne Baur is supported by CERFACS through the project MIRAGE. BS and RS acknowledges funding by the European Union's Horizon 2020 (H2020) research and innovation program under Grant Agreement No. 101003536 (ESM2025 – Earth System Models for the Future), 821003 (4C, Climate-Carbon Interactions in the Coming Century) and 101003687 (PROVIDE).

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
