# Peer review of "Solar Radiation Modification challenges decarbonization with renewable solar energy"

_EGUsphere, 2023_

## Referee Comment (RC2)

**General comment:**

This manuscript analyzed G6sulfur using CNRM-ESM2-1 to estimate the impact of injected sulfate aerosol on two types of renewable energy: photovoltaic (PV) and concentrated solar power (CSP). It concluded that solar power production potential would decrease under G6sulfur relative to SSP8-8.5 and SSP2-4.5, with CSP efficiency experiencing a greater reduction than PV. This reduction is mainly attributed to the significant reduction in direct solar radiation, which strongly affect the efficiency of CSP, while the increase of diffuse radiation under G6sulfur slightly improves PV efficiency. For both CSP and PV, the cooling effect and reduced cloud coverage under G6sulfur have a positive influence on solar power production. This study is important to fill the gap in understanding how solar radiation modification impacts renewable energy and is a good fit for Earth System Dynamics.

However, there are a few issues that need to be addressed before publication.
1. Please clarify whether you have repeated G6sulfur using CNRM-ESM2-1 with an updated aerosol-light interaction. If so, clearly indicate the difference between the previous G6sulfur simulation and the current one. It might be worth to compare the radiation changes in the two sets of G6sulfur. Also please make those output available.
2. Please explain how the individual forcing of cloudy sky and clear sky was calculated. As far as I know the model output of cloudy sky includes both the effect of aerosol and cloud. How did you separate the surface radiation effect from cloud and aerosol?
3. Correct the supplement figure citations in the manuscript. The color bars in Figure 5 and Figure S11 need to be reversed.

**Specific comments:**

Line 15: "simulated 1-hour output…are used for the assessment" is not clear what SAI and reference future scenarios are used. It needs to be clarified that SAI is applied under SSP585 to reduce the radiative forcing from SSP585 to SSP245. This is GeoMIP G6sulfur experiment.

Line 18-20: clarify this sentence.

Line 25-27: citation of (de Coninck et al., 2018) right after 'solar geoengineering' confuses readers that this citation is for solar geoengineering, but instead, this is for 1.5 oC temperature target.

Line 51-63: what is the global warming scenario used in this paragraph. I would guess different SSPs have different changes.

Line 86: there are only three different scenarios, not four. SSP2-4.5, SSP5-8.5, G6sulfur

Line 92: "accounts for the aerosol-light interaction". Does this mean diffuse radiation increases after SAI? How about aerosol-light interaction in the troposphere? The G6sulfur output on Earth System Grid from CNRM-ESM2-1 are showing there is less diffuse radiation in G6sulfur

than SSP5-8.5. And the reason is that CNRM-ESM2-1 does not count for the aerosol scattering effect. Since G6sulfur has less cloud coverage than SSP5-8.5, there is less diffuse radiation in G6sulfur. But if the scattering effect is considered, there should be more diffuse radiation in G6sulfur than in SSP5-8.5 (as other models show in G6sulfur). If the authors have re-run G6 simulations, it would be valuable to upload those output on ESG labeled with G6.

Line 106: what is STC?

Equation 1: just to confirm that PV-TP takes direct and diffuse radiation with the same efficiency, since RSDS is used. Does PV panel need to adjust certain setting for better performance under cloudy days?

Equation 3: should c3 be different for direct and diffuse radiation?

Figure S4 and S5 are confusing. Captions of Fig. S4 and S5 are the same. Shouldn't the two panels of figures together be Figure S4?

What is the difference between Figure 2 and Figure S4, S5? They seem like to plot the same variables, but the values are different. Which ones are correct?

Figure 2: PV and CSP are using different scales in the color bars. Please use the same scale for better comparison. The subtitle cannot describe how the values are calculated. It might be better to use '(SAI minus ssp245)/ssp245'? This also applies to other maps later.

Line 205-214: Please be consistent with the format when referring to different panels in one figure. This paragraph used 'Fig. 3a-c', 'Fig.3 d-f', 'Fig. g-I'.

Figure 3: still the subtitle cannot reflect how the values are calculated.

Line 209: please clarify 'cloudy sky' only consider cloud effect on radiation. Also, how was this calculated? The model also output radiation without aerosol effect under SAI?

Line 220: it should be Figure 4a instead of 3a.

Figure 4: How well does CNRM-ESM2-1 simulate dust? Over Sahara region, dust emission and concentration may play an important role.

Figure 4: it seems that PV reduction in winter (DJF in northern hemisphere, JJA in southern hemisphere) is stronger than in summer. What is the reason?

Figure 5: In previous figures, purple was used for negative changes, and green was used for positive. But here the color code is opposite. Please keep it consistent.

Line 322-327: it is important to mention the SAI strategy and climate model used when comparing this study with others.

Line 364: This conclusion cannot draw from Fig. S4, S5, and S13.

Conclusion: It is not necessary to redefine SRM, SAI, PV and CSP.

Code and data availability statement:
The new climate model simulation should be available to public.

---

## Author Comment (AC1)

Reviewer #1

Baur et al. evaluate the potential for solar photovoltaic and concentrating solar power under three future climate scenarios: SSP5-8.5, SSP2-4.5, and G6sulfur which reduces the climate forcing from SSP5-8.5 to SSP2-4.5 using stratospheric aerosol injection (SAI). They find the resource potential for both technologies reduces under the geoengineering scenario. The results confirm the one study that has previously investigated SAI impacts on solar energy technologies, from Smith et al. (2017).

The study is a development over Smith et al. (2017) in two regards. Firstly, the authors consider locational feasibility of solar power installations, ruling out or downweighting grid cells that are in protected areas, far from population centres, and conflict with existing land use types. The second is that the authors consider the intra-year variability in solar energy resource, referring to "low energy weeks" in which meteorological conditions do not produce sufficient energy. I also quite like that the authors used hourly data output from the climate model (compared to three-hourly from Smith et al.). With these additions, the results are similar to Smith et al. (2017), indicating robustness in the (admittedly intuitive) statement that SAI leads to reduced solar energy potential. Given the increasing occurrence of SAI in policy discourse, it is important that studies like these get a renewed focus, as the negative impacts on conventional mitigation (e.g. renewable technologies) of geoengineering are often not considered.

We appreciate the reviewer's comments and are pleased to hear that they find the study to be a valuable contribution to the SRM discourse.

Main comments:

1. There does not appear to be a consideration of the solar geometry in the equation for PV. For CSP, the factor of the cosine of the zenith angle cancels out (Smith et al. eq. (7), eq. (9)) so providing the FLH equation is correctly defined in your paper then this is OK. However for PV, the direct/total irradiance is important, as well as the orientation of the solar panel, as to the amount of radiation it receives and the panel temperature which affects its efficiency. In eq. (1), the power output expected would be greater than predicted from the climate model value of RSDS, since this would be a horizontal irradiance value, and a real-world solar plant operator would angle the panels appropriately to maximise the incident irradiance on the panel. Perhaps these corrections are already baked into the equations you use. It would be good to confirm.

   We agree that the consideration of the solar geometry and the exposition of the panels is a better representation of real-world solar farm installations. In our revised manuscript we added another subchapter to the methods and results showing PV potential when solar geometry and panel inclination are accounted

for and discuss these additional results in the following parts of the paper. Unfortunately, we could not update the entire analysis of our study because a processed decomposition of the single physical drivers (Figure 3) is not possible for us under separation of direct and diffuse light (the model doesn't produce the RSDSdiff-clear-sky variable as a standard output).

The main conclusion of the paper does not change with this updated version of the PV potential calculation but it shows even larger relative and absolute reductions in the high latitudes than for horizontally aligned panels (Figure 1). This is because the tilt of the panel increases the amount of direct radiation that can be harvested. However, SAI modifies the fraction of direct and diffuse radiation to entail a larger diffuse fraction and therefore the advantage of the tilt is reduced under SAI versus the SSP scenarios. Hence, relative reductions in high latitudes under SAI that already exist for horizontally aligned panels are further increased for tilted panels.

[Figure]

*Figure 1: Difference in 2090-2099 PV potential with fixed tilted panels between the ensemble means of SAI and a) SSP245, b) SSP585 and c) absolute difference between latitudinal zonal sums between SAI and SSP245 and SSP585 in PWh/year. White areas have a SNR of < 1. x -> y denotes (y − x)/x.*

We added a figure to the Supplementary Information to illustrate the effect of the tilt and solar geometry on the direct and diffuse radiation that reaches the panels surface (Figure 2).

[Figure]

*Figure 2: Difference in the direct and diffuse components of the PV potential calculation when solar geometry and panel tilt are accounted for (RSDSpanel) versus when radiation on a horizontally aligned panel is considered (RSDS). a-c) display the difference in diffuse radiation that is used in RSDSpanel versus in RSDS. d-f) same as a-c but for direct radiation. g-i) same as a-c but for total radiation.*

2. Around line 272 there is a "quasi-linear" relationship for reduced potential. It looks fairly linear in time (fig. 6), but since we don't have the SAOD plot we don't know if it is linear in AOD. This would be quite a useful result to verify as if it is linear, it would be easy to transplant into an economic or integrated assessment model.

We agree and added the 10-year global mean AOD to Figure 7 (in the manuscript; Figure 3 in this response).

[Figure]

*Figure 3: Relative difference over time of SAI (red), SSP245 (gray) and SSP585 (black) PV potential compared to 2015-2024 values for selected regions and global aerosol optical depth times -1 (blue) to compare change in PV potential with the magnitude of SAI deployment. Lines are the ensemble means with the bars indicating the 20-80 percentile ranges of the single members. X -> y denotes (y – x)/x.*

*"Figure 7 illustrates the temporal evolution of the relative difference between the three scenarios and present-day values with the increase in SAI deployment intensity over time. In the first four decades, the scenarios differ only slightly, but the gap in potential starts to widen as time goes on. The quasi-linear increase in the gap between SAI and the SSP-scenarios in some regions indicates that, in these areas, the reduction in PV potential strengthens with increasing global mean aerosol optical depth."*

Minor points

Lines 35-37: several of the references are repeated

Done.

Line 73: "dystopian" I suppose is a slight value judgement

We removed it.

Line 86: A brief descripton of what G6sulfur aims to do, and the experiment design, would be useful.
Line 87: "imitates": I take from this that CNRM-ESM is not emissions driven for stratospheric aerosol injection. It is mentioned in the discussion, but would be good

to introduce here. Related to my comment about experiment design, what is the total loading or optical depth required to achieve the desired avoided warming?

We already give a brief description of what G6sulfur aims to do and refer to Ben Kravitz et al., 2015 for more information. However, we have included a sentence on the total aerosol optical depth (which is now also displayed in Figure 7 in the manuscript) and the fact that the scenarios are run in concentration-driven mode already in the Methods chapter. We also added more details on the difference between the existing CNRM-ESM2-1 G6sulfur simulations and the repetition of the simulations we have performed for this paper.

*"We calculate the potential for three different scenarios: SSP245, a scenario representing approximate current policy (O'Neill et al., 2016), SSP585, a very high-emission scenario (O'Neill et al., 2016), and G6sulfur, an SRM scenario that imitates stratospheric aerosol injections (SAI) (Kravitz et al., 2015) and will be referred to as SAI in this study. G6sulfur has the initial conditions and underlying emissions of SSP585 but uses SAI to match the global radiative balance of SSP245 until 2100. G6sulfur is part of the GeoMIP protocol (Kravitz et al., 2015), but here, the setup is enhanced with higher frequency output and additional variables related to radiation and wind. We run the scenarios using the Earth System Model CNRM-ESM2-1 with prescribed aerosol optical depth derived from the GeoMIP experiment G4SSA (Tilmes et al., 2015) to simulate the aerosol injections in G6sulfur/SAI. 3-member ensembles of G6sulfur/SAI, SSP245 and SSP585 from CNRM-ESM2-1 exist already, but are not used here. Instead, for this study, we repeated the simulations with an alternative version of CNRM-ESM2-1 (Séférian et al., 2019) that accounts for the aerosol-light interaction. This additional feature of the model enables a change in the partition of direct and diffuse light due to a change in aerosol concentration in the whole atmospheric column. We run a 6-member ensemble with initial condition perturbations as for the standard SSP-simulations for all three scenarios in concentration-driven mode. The simulations cover the 2015-2100 period and output data is saved at hourly frequency. The global mean aerosol optical depth required in the SAI simulation to get from SSP585 to SSP245 reaches 0.35 in the last decade."*

Line 92: "aerosol-light interaction", do you mean "aerosol-radiation interaction"?

Yes.

Equations 1 and 4: the LHS looks like a subtraction, would be better to be a subscript Tpi

Done.

Line 108 & 133: Format -2 superscript

Done.

Line 134, and a few other places – reference formatting a little sloppy and haphazard

We corrected the in-line reference formatting.

Line 157: is there a basis for choosing 500 km as the cut-off?

No, there isn't. We added:
*"until it reaches a weight of 0 at a distance of about 500 km, an arbitrarily chosen cut-off."*

Line 169: are low energy week boundaries fixed (i.e. Monday to Sunday), or do you take 7-day rolling averages?

The boundaries are fixed because a counting of weeks would not be possible with rolling sums. We included a sentence in the description of the LEW metric:

*"The boundaries of the 7-day period are fixed."*

Line 183: do the different population masks significantly affect the results? It feels like this isn't quite an apples to apples comparison.

Yes, different population masks affect which areas are considered suitable for solar farms and significantly affect the results (see for example S4d-f). For this to be as close to an apples-to-apples comparison we chose the same population setting for all scenarios.

Line 208: delete first "in"

Great, thanks.

Line 275: "a lot less well-behaved" – being a bit more explicit/formal here is useful.

We have included this sentence:

*"The temporal evolutions are a lot less well-behaved with relative increases and decreases compared to the present that can be several times larger when the land-use suitability area weighting is included respective to its scenarios (Fig. S14) than when it is not (Fig. S13)."*

---

## Author Comment (AC2)

Reviewer #2
**General comment:**
This manuscript analyzed G6sulfur using CNRM-ESM2-1 to estimate the impact of injected sulfate aerosol on two types of renewable energy: photovoltaic (PV) and concentrated solar power (CSP). It concluded that solar power production potential would decrease under G6sulfur relative to SSP8-8.5 and SSP2-4.5, with CSP efficiency experiencing a greater reduction than PV. This reduction is mainly attributed to the significant reduction in direct solar radiation, which strongly affect the efficiency of CSP, while the increase of diffuse radiation under G6sulfur slightly improves PV efficiency. For both CSP and PV, the cooling effect and reduced cloud coverage under G6sulfur have a positive influence on solar power production. This study is important to fill the gap in understanding how solar radiation modification impacts renewable energy and is a good fit for Earth System Dynamics.

We appreciate the reviewer for taking the time to provide constructive feedback on our manuscript and for considering it suitable for ESD.

However, there are a few issues that need to be addressed before publication.
1. Please clarify whether you have repeated G6sulfur using CNRM-ESM2-1 with an updated aerosol-light interaction. If so, clearly indicate the difference between the previous G6sulfur simulation and the current one. It might be worth to compare the radiation changes in the two sets of G6sulfur. Also please make those output available.
   We have expanded the description of the G6sulfur experiments and the difference between existing CNRM-ESM2-1 G6sulfur simulations and the new ones we have run. Of course, the outputs will be made available after publication.
   *"We calculate the potential for three different scenarios: SSP245, a scenario representing approximate current policy (O'Neill et al., 2016), SSP585, a very high-emission scenario (O'Neill et al., 2016), and G6sulfur, an SRM scenario that imitates stratospheric aerosol injections (SAI) (Kravitz et al., 2015) and will be referred to as SAI in this study. G6sulfur has the initial conditions and underlying emissions of SSP585 but uses SAI to match the global radiative balance of SSP245 until 2100. G6sulfur is part of the GeoMIP protocol (Kravitz et al., 2015), but here, the setup is enhanced with higher frequency output and additional variables related to radiation and wind. We run the scenarios using the Earth System Model CNRM-ESM2-1 with prescribed aerosol optical depth derived from the GeoMIP experiment G4SSA (Tilmes et al., 2015) to simulate the aerosol injections in G6sulfur/SAI. 3-member ensembles of G6sulfur/SAI, SSP245 and SSP585 from CNRM-ESM2-1 exist already, but are not used here. Instead, for this study, we repeated the simulations with an alternative version of CNRM-ESM2-1 (Séférian et al., 2019) that accounts for the aerosol-light interaction. This additional feature of the model enables a change in the partition of direct and diffuse light due to a change in aerosol concentration in the whole atmospheric column. We run a 6-member ensemble with initial condition perturbations as for the standard SSP-simulations for all three scenarios in concentration-driven mode. The simulations cover the 2015-2100 period and output data is saved at hourly frequency. The global mean aerosol optical depth required in the SAI simulation to get from SSP585 to SSP245 reaches 0.35 in the last decade."*

2. Please explain how the individual forcing of cloudy sky and clear sky was calculated. As far as I know the model output of cloudy sky includes both the effect of aerosol and cloud. How did you separate the surface radiation effect from cloud and aerosol?
We added these sentences for clarification to the Methods:
*"The calculation of the cloudy sky radiation involves the subtraction of clear sky downwelling shortwave radiation from the total downwelling shortwave radiation. The clear sky downwelling shortwave radiation is a variable that excludes the effect of clouds but includes aerosols, and is saved by the model."*

3. Correct the supplement figure citations in the manuscript. The color bars in Figure 5 and Figure S11 need to be reversed.
We purposefully reversed the color bars for the LEW figures since a reduction here is a "positive" result in terms of energy production. In the revised manuscript we will either explicitly note the inversion of the color bar or use different colors entirely.

**Specific comments:**
Line 15: "simulated 1-hour output...are used for the assessment" is not clear what SAI and reference future scenarios are used. It needs to be clarified that SAI is applied under SSP585 to reduce the radiative forcing from SSP585 to SSP245. This is GeoMIP G6sulfur experiment.
We clarified that the SAI experiment involves cooling down from the ssp585 baseline to ssp245 in the abstract. To avoid complexity, however, we limit the use of abbreviations in the abstract and remark in the main text that SAI = G6sulfur.
*"The SRM scenario uses Stratospheric Aerosol Injections (SAI) to approximately lower global mean temperature from the unmitigated, high-emission scenario SSP585 baseline to a moderately mitigated scenario SSP245."*

Line 18-20: clarify this sentence.
*"We find that by the end of the century, most regions experience an increased number of low PV and CSP energy weeks per year under SAI compared to SSP245. Compared to SSP585, while the increase in low energy weeks under SAI is still dominant on a global scale, certain areas may benefit from SAI and see fewer low PV or CSP energy weeks. A substantial part of the decrease in potential with SAI compared to the SSP-scenarios is compensated by optically thinner upper tropospheric clouds under SAI which allow more radiation to penetrate towards the surface."*

Line 25-27: citation of (de Coninck et al., 2018) right after 'solar geoengineering' confuses readers that this citation is for solar geoengineering, but instead, this is for 1.5 oC temperature target.
We have clarified the citation situation:
*"With a rapidly dwindling remaining carbon budget for the Paris 1.5 °C temperature goal, a growing set of literature has been investigating the potential of temporarily reducing climate change impacts with Solar Radiation Modification (SRM), also known as solar geoengineering (UNEP, 2023)."*

Line 51-63: what is the global warming scenario used in this paragraph. I would guess different SSPs have different changes.

Yes, you are right. Although currently the uncertainties are quite large and agreement on the sign of change regardless of the magnitude of the climate forcing is already a success. We have removed any ranges of change and now refer only to increases and decreases, which slightly vary depending on study, model and underlying forcing magnitude.

Line 86: there are only three different scenarios, not four. SSP2-4.5, SSP5-8.5, G6sulfur

Yes, we corrected it.

Line 92: "accounts for the aerosol-light interaction". Does this mean diffuse radiation increases after SAI? How about aerosol-light interaction in the troposphere? The G6sulfur output on Earth System Grid from CNRM-ESM2-1 are showing there is less diffuse radiation in G6sulfur than SSP5-8.5. And the reason is that CNRM-ESM2-1 does not count for the aerosol scattering effect. Since G6sulfur has less cloud coverage than SSP5-8.5, there is less diffuse radiation in G6sulfur. But if the scattering effect is considered, there should be more diffuse radiation in G6sulfur than in SSP5-8.5 (as other models show in G6sulfur). If the authors have re-run G6 simulations, it would be valuable to upload those output on ESG labeled with G6.

Yes, in these new simulations the model does account for the aerosol scattering effect in the stratosphere and troposphere and there is more diffuse radiation under G6sulfur than ssp585. We are still looking into finding a solution to publicly provide the raw data. Due to the hourly output we need a public repository where ~1.5TB can be stored. 1.5TB would mean 1 member, only the 6 variables of interest for this study and only the last decade of the century. If the reviewer has any suggestions on where this data can be stored without high fees we would much appreciate any advice. Otherwise, we would upload 10-year means instead of hourly output on zenodo with a DOI.

Line 106: what is STC?

Standard Test Conditions. It is explained further below, we corrected it, thanks for pointing it out.

Equation 1: just to confirm that PV-TP takes direct and diffuse radiation with the same efficiency, since RSDS is used. Does PV panel need to adjust certain setting for better performance under cloudy days?

Equation 3: should c3 be different for direct and diffuse radiation?

Yes, PV-TP takes direct and diffuse radiation with the same efficiency. On cloudy days panels would likely adjust the tilt of the panel.

We have added another mode of PV-TP calculation (PV-TP$_{fixed}$) to the study that takes direct and diffuse radiation into account separately by considering solar geometry and a tilt of the panel. The main conclusion of the paper does not change with this updated version of the PV potential calculation but it shows even larger relative and absolute reductions in the high latitudes than for horizontally aligned panels (Figure 1). This is because the tilt of the panel increases the amount of direct radiation that can be harvested. However, SAI modifies the fraction of direct and diffuse radiation to entail a larger diffuse fraction and therefore the advantage of the tilt is reduced under SAI versus the SSP scenarios. Hence, relative reductions in

high latitudes under SAI that already exist for horizontally aligned panels are further increased for tilted panels.

[Figure]

*Figure 1: Difference in 2090-2099 PV potential with fixed tilted panels between the ensemble means of SAI and a) SSP245, b) SSP585 and c) absolute difference between latitudinal zonal sums between SAI and SSP245 and SSP585 in PWh/year. White areas have a SNR of < 1. x -> y denotes (y – x)/x.*

We added a figure to the Supplementary Information to illustrate the effect of the tilt and solar geometry on the direct and diffuse radiation that reaches the panels surface (Figure 2).

[Figure]

*Figure 2: Difference in the direct and diffuse components of the PV potential calculation when solar geometry and panel tilt are accounted for (RSDSpanel) versus when radiation on a horizontally aligned panel is considered (RSDS). a-c) display the difference in diffuse radiation that is used in RSDSpanel versus in RSDS. d-f) same as a-c but for direct radiation. g-i) same as g-i but for total radiation.*

Figure S4 and S5 are confusing. Captions of Fig. S4 and S5 are the same. Shouldn't the two panels of figures together be Figure S4? What is the difference between Figure 2 and Figure S4, S5? They seem like to plot the same variables, but the values are different. Which ones are correct?

As suggested, we have turned the two figures into one with two panels. We have also modified the caption to be clearer on what is the difference between the two. The difference is the land use suitability and population density assumptions. Figure 2 uses the same assumptions for all 3 scenarios. Figure S4 a-c uses land use suitability assumptions that are scenario-specific and S4 d-f uses land use suitability and population density assumptions that are scenario-specific.

Figure 2: PV and CSP are using different scales in the color bars. Please use the same scale for better comparison. The subtitle cannot describe how the values are calculated. It might be better to use '(SAI minus ssp245)/ssp245'? This also applies to other maps later.
We have adjusted the scales to be the same for PV and CSP and included a clearer description of the calculation as you have laid out in the captions of the relevant figures to keep readability of the plots.

Line 205-214: Please be consistent with the format when referring to different panels in one figure. This paragraph used 'Fig. 3a-c', 'Fig.3 d-f', 'Fig. g-l'.
Done.

Figure 3: still the subtitle cannot reflect how the values are calculated.
Line 209: please clarify 'cloudy sky' only consider cloud effect on radiation. Also, how was this calculated? The model also output radiation without aerosol effect under SAI?
Regarding cloudy and clear sky we have added this to the Methods:
*"The calculation of the cloudy sky radiation involves the subtraction of clear sky downwelling shortwave radiation from the total downwelling shortwave radiation. The clear sky downwelling shortwave radiation is a variable that excludes the effect of clouds but includes aerosols, and is saved by the model."*
And this to the caption of Figure 3:
*"a-c) was calculated by keeping all variables except temperature fixed, d-f) by keeping all variables except radiation fixed, g-i) by using the model output clear sky radiation instead of total radiation (see Methods) and j-l) by subtracting clear sky radiation from total radiation."*

Line 220: it should be Figure 4a instead of 3a.
Yes, thank you.

Figure 4: How well does CNRM-ESM2-1 simulate dust? Over Sahara region, dust emission and concentration may play an important role.
CNRM-ESM2-1 simulates dust well, especially over the Sahara and the Middle East. However, the total optical thickness of dust is slightly underestimated compared with satellite estimates. For detailed dust assessments there's Checa-Garcia et al., 2021 (https://doi.org/10.5194/acp-21-10295-2021) (the 3DU version) and Michou et al., 2019 (https://doi.org/10.1029/2019MS001816). The effect of dust on PV and CSP potential should therefore be fairly well represented in our analysis. It could explain why deserted areas see a comparably high LEW number, although that remains speculative at this point.

Figure 4: it seems that PV reduction in winter (DJF in northern hemisphere, JJA in southern hemisphere) is stronger than in summer. What is the reason?

The reduction in *absolute* potential is higher in summer, of course. Our hypothesis is that the *relative* reduction is larger in winter because the sun is at a higher zenith angle and therefore has to pass through a greater layer of aerosol optical depth.

Figure 5: In previous figures, purple was used for negative changes, and green was used for positive. But here the color code is opposite. Please keep it consistent.
Re response to 3. above.

Line 322-327: it is important to mention the SAI strategy and climate model used when comparing this study with others.
We enhanced the sentences with the following simulation and model information: *"Visioni et al. (2021) illustrate the global pattern of total cloud cover differences between a SAI simulation that compensates for the temperature increase above present values under RCP8.5 conditions and the control run close to the present period. The simulations were performed with the CESM-WACCM model. Not only did they find…"*

Line 364: This conclusion cannot draw from Fig. S4, S5, and S13.
We are convinced that we can. Especially looking at the absolute changes (Fig S4c,f) (former Figs S4 and S5) and regional relative changes S12 (former Fig S13).

Conclusion: It is not necessary to redefine SRM, SAI, PV and CSP.
Ok, we removed the redefinitions.

Code and data availability statement:
The new climate model simulation should be available to public.
See response above. If a large enough repository cannot be found we will upload the 10-year means on zenodo.